# *Clostridioides difficile* Infection in Hospitalized Patients—A Retrospective Epidemiological Study

**DOI:** 10.3390/healthcare12010076

**Published:** 2023-12-29

**Authors:** Frederico Fonseca, Mario Forrester, Ana Margarida Advinha, Adriana Coutinho, Nuno Landeira, Maria Pereira

**Affiliations:** 1Pharmaceutical Services, Hospital do Espírito Santo, 7000-811 Évora, Portugal; nlandeira@hevora.min-saude.pt (N.L.); dir.farmacia@hevora.min-saude.pt (M.P.); 2Sociedade Portuguesa dos Farmacêuticos dos Cuidados de Saúde, 3030-320 Coimbra, Portugal; mario.forrester.quesada@ubi.pt; 3Faculty of Health Sciences, UBI—Universidade da Beira Interior, 6200-506 Covilhã, Portugal; 4UFUP—Unidade de Farmacovigilância da Universidade do Porto, 4200-450 Porto, Portugal; 5CHRC—Comprehensive Health Research Centre, University of Evora, 7000-811 Évora, Portugal; anamma@uevora.pt; 6Department of Health and Medical Sciences, School of Health and Human Development, University of Evora, 7000-671 Évora, Portugal; 7Laboratory Services, Microbiology Department, Hospital do Espírito Santo, 7000-811 Évora, Portugal; acoutinho@hevora.min-saude.pt

**Keywords:** *Clostridioides difficile* infection, diarrhea, epidemiology, healthcare-acquired, risk factors, antibiotic consumption, community-acquired

## Abstract

*Clostridioides difficile* infection (CDI) is the main source of healthcare and antibiotic-associated diarrhea in hospital context and long-term care units, showing significant morbidity and mortality. This study aimed to analyze the epidemiological context, describing the severity and outcomes of this event in patients admitted to our hospital, thus confirming the changing global epidemiological trends in comparison with other cohorts. We conducted a single-center, observational, and retrospective study at the Hospital do Espírito Santo (HESE), Évora, in Portugal, analyzing the incidence of CDI in patients meeting eligibility criteria from January to December 2018. During this period, an annual incidence rate of 20.7 cases per 10,000 patients was documented. The studied population average age was 76.4 ± 12.9 years, 83.3% over 65. Most episodes were healthcare-acquired, all occurring in patients presenting multiple risk factors, with recent antibiotic consumption being the most common. Regarding severity, 23.3% of cases were classified as severe episodes. Recurrences affected 16.7% of participants, predominantly female patients over 80 years old, all of whom were healthcare-acquired. Mortality rate was disproportionately high among the older population. Our investigation documented an overall incidence rate of over 10.4-fold the number of cases identified in the year 2000 at the same hospital, more recently and drastically, in community-associated episodes.

## 1. Introduction

*Clostridioides difficile* (CD) is spore forming, gram-positive anaerobic bacterium, capable of infecting the gastrointestinal tract, responsible for CDI, through the release of enterotoxin A and cytotoxin B, causing a diverse spectrum of conditions, varying from asymptomatic colonization or mild diarrhea to fulminant life-threatening colitis [1,2]. This is the most frequently diagnosed cause of antimicrobial and healthcare-associated diarrhea in hospital institutions and long-term care units [3,4]. Over the past decades, numerous studies have reported a significant increase in the incidence and severity of healthcare-associated CDI, and more recently community-acquired episodes. This has been associated with certain hypervirulent strains, specifically ribotype 027 (BI/NAP1/027), responsible for producing a markedly increased toxins level, leading to high morbidity and mortality [5]. A survey funded by the European Center for Disease Prevention and Control (ECDC) in 2008 reported a 71% increase in CDI incidence compared to previous European surveillance studies [6]. As a result, the Centers for Disease Control and Prevention and other institutions have made significant efforts to prevent infections and implement epidemiological surveillance, resulting in a decrease in CD standardized infection ratio and its disease burden [7]. This trend in incidence may partially be explained by changes in high-risk antibiotic use. Inappropriate, long term and cumulative exposure to clindamycin, penicillin, quinolones and carbapenems has become a well-established risk factor for developing CDI, as it disrupts the intestinal microbiota, promoting the overgrowth of CD, therefore increasing the risk of infection until three months after its cessation [8,9,10,11,12]. In primary care settings, it has been found that at least 20% of prescribed antibiotics are deemed unnecessary [13]. Therefore, the implementation of Antimicrobial Stewardship Programs are priority interventions in the control of healthcare-associated infections [14]. During the period of 2012 to 2021, the ECDC documented a significant decrease in total consumption of antibacterials, observed for the EU overall [15]. There are additional risk factors that predispose patients to CDI, including advanced age (over 65 years), recent hospitalization or long length of stay in a healthcare setting, recent gastrointestinal surgery and the use of specific medications, such as proton pump inhibitors (PPIs), non-steroidal anti-inflammatory drugs (NSAIDs) and immunosuppressants. Patients receiving acid suppressing agents have a higher probability of developing CDI, due to disruption of the indigenous gut microbiota [16,17].

The Infectious Diseases Society of America (IDSA), the Society for Healthcare Epidemiology of America (SHEA) and the European Society of Clinical Microbiology and Infectious Diseases (ESCMID) have published the most updated guidelines on CDI management [18,19,20]. Developed by multidisciplinary specialists in infection prevention and control, these recommendations prioritize discontinuing unnecessary antibiotic therapy and ensuring proper fluid and electrolyte replacement as the initial therapeutic approach [18,19,20]. In 2017, these guidelines brought important adjustments in CDI management, resulting in dynamic changes in treatment orientations, currently suggesting either fidaxomicin or vancomycin as preferred choices over metronidazole, due to recent data proven inferiority [19,20]. As a result, metronidazole is no longer recommended as a first-line antibiotic agent and should only be considered when fidaxomicin and vancomycin are not feasible or available. Fidaxomicin now is the preferred option due to its lower recurrence rate and higher efficiency in eradicating CD [20]. In 2021, the ESCMID supported by a literature review performed by the Study Group for CD, published a treatment guidance document, discussing novel approaches, such as fecal microbiota transplantation and toxin-binding monoclonal antibodies, highlighting their high efficacy and safety profiles [20,21]. Bezlotoxumab was the first humanized monoclonal antibody against CD toxin B approved for the prevention of recurrent disease in conjunction with standard-of-care (SoC) antibiotic antibiotics by the U.S. Food and Drug Administration in October 2016 and afterwards by the European Medicines Agency in January 2017 [22]. This approval was supported by phase 3 clinical trials data, showing reduced rates of CDI recurrence as an adjuvant treatment compared with placebo [23]. Moreover, a study conducted in Portugal, known as PIN35, suggests that bezlotoxumab in addition to SoC treatment had a positive cost-effective ratio, reducing the disease recurrence rate, and economic burden from medical costs and indirect expenditures, associated with prolonged length of stay [24].

Supportive care is a crucial aspect of managing CDI, alongside antibiotic treatment. It involves a multidisciplinary approach, preventing electrolyte imbalance, dehydration, or malnutrition. Patients should also be isolated to prevent infection transmission. Other strategies, such as probiotics use, which are microorganisms that reinforce the epithelial barrier by competitively excluding pathogenic bacteria from adhering to the intestinal mucosa, offer potential health benefits for both CDI prevention and treatment [25,26]. However there has been limited evidence to support their effectiveness and future research and stronger evidence will be needed to elucidate and evaluate the most effective probiotics, optimum amount and ideal duration, as adjuvant therapy with SoC antibiotics. Studies estimate a recurrence rate range from 10 to 30% of all cases, and published data suggests that second and/or subsequent episodes lead to not only significant increase in morbidity and mortality, but also increased medical costs to the National Health Service (NHS) [5]. These financial implications transcend healthcare institutions implementing strict infection control measures, such as environmental cleaning, including indirect medical costs and cover productivity losses attributed to longer hospital stays and work absenteeism. This significantly contributes to the overall financial burden posed by CDI, adding to the overall financial burden [27]. As a result, other strategies and alternative therapies have been explored and identified, leading to the discovery of new and more effective drugs [21,22]. 

One of the main objectives and activities foreseen for the Portuguese NHS in 2018, according to the Ministry of Health, was to expand the epidemiological surveillance of CDI [28]. Accordingly, our study gives a comprehensive perspective on the current paradigm of this event in Portugal. Specifically, we perform an epidemiological analysis of this infection on hospitalized patients at HESE, in Portugal, during 2018.

## 2. Materials and Methods

### 2.1. Study Design

We conducted a single-center, observational and retrospective study using patients with CDI diagnosis at HESE, between January and December 2018. All episodes and related recurrences occurring during the study period were identified and recorded for patients meeting eligibility criteria. All studied subjects had a confirmed diagnosis of CDI, presenting diarrhea, defined as three or more unformed stools in 24 h, and positive stool test for toxigenic CD and/or its toxins, or histological diagnosis (pseudomembranous colitis). The primary endpoint was to evaluate the overall incidence of CDI episodes as the average annual incidence (number of cases per 10,000 admissions), over the course of a 1-year period. Furthermore, we aimed to analyze the correlation between pre-exposure antibiotic drugs and other risk factors with the severity of infection, as well as determine the effectiveness of different treatment regimens. 

Additionally, we perform a supplementary descriptive review, focusing only on infection overall incidence, and both healthcare and community-acquired prevalence, during an extended four-year period, from 2019 to 2022, using patients’ sociodemographic data, meeting the same eligibility criteria and case origin information. The purpose of this extended period analysis was to conduct a comparative analysis, evaluating the alignment between the epidemiological changes documented in the existing literature and the findings derived from our study.

### 2.2. Inclusion and Exclusion Criteria

The subjects meeting eligibility criteria were all adults (≥18 years), institutionalized at our hospital facilities, who had a confirmed diagnosis of CDI, according to the ECDC surveillance protocol and met at least one of the following criteria: diarrheal stools with a positive laboratory assay for CD toxin A and/or B or histological diagnosis, presenting pseudomembranous colitis revealed by colonoscopy, between January and December 2018 [29]. Other causes of diarrhea, unrelated to CD, with positive results for other bacteria such as *Salmonella* spp., *Shigella* spp., *Escherichia coli* or viruses, such as *Rotavirus*, were excluded from the study. Any fecal specimen sample not fulfilling clinical guidance standard Norms of Clinical Orientation, guideline No. 019/2014 submission criteria, described by the Portuguese Directorate-General of Health, such as formed stool specimens, samples stored above the 2–8 °C temperature range until delivery time in laboratory services, microbiology department, or samples exceeding 24 to 48 h after collection, were also excluded from the study [30].

### 2.3. Data Collection and Study Population

Digital data were collected from all hospitalized patients at our institution facilities who were diagnosed with CDI during the study period and met the eligibility criteria. Sociodemographic (age group and sex), epidemiological (origin of acquisition), underlying conditions or comorbidities of interest (diabetes, cardiovascular disease, kidney dysfunction, obesity, cancer, recent history of surgical interventions), laboratory test results, clinical and therapeutic data were documented. The Charlson Comorbidity Index (CCI) score was used as a method of categorizing comorbidities in patients, based on the International Classification of Diseases diagnosis codes found in our hospital records data [31]. Additionally, risk factors such as recent antibiotic consumption, advanced age and hospitalization until eight weeks before disease onset, were collected. Patients’ information, including medical records, clinical reports, laboratory results and drug prescription records were accessed and recorded using digital systems, such as ALERT^®^, v2.8.3.1 and Sclinico^®^, v2.8, Electronic Medical Prescription (PEM^®^ v2.4.0) system and Hospital Information System, (HS-SGICM—Glintt^®^, v2.0). We conducted a descriptive review, comparing our research with other epidemiological studies, extracting crucial details like the first author’s name, period of study, research location, overall CDI incidence, and antibiotic exposure rates for comparative purposes. Employing a comprehensive and structured approach, we crafted a search strategy to identify pertinent studies using specific keywords in databases such as PubMed and Scopus and selected studies based on their epidemiological relevance, aligning inclusion criteria with our study, ensuring quality assessment. We extracted and compared CDI incidence rates, measured as the number of cases per 10,000 admissions, depending on the defined cases.

### 2.4. Cases Identification/Classification and Definitions

According to ECDC and IDSA/SHEA, a primary incident case of CDI is defined by the presence of diarrhea and a positive stool test (toxin or molecular assay) from a subject with no positive test in the prior eight weeks. There are several classifications based on disease origin/epidemiology and severity. According to these guidelines we established the CDI origin in our study as healthcare-acquired (HA-CDI) or community-acquired (CA-CDI) [18,32]. Patients with positive samples obtained more than 48 h after admission at the hospital were classified as HA-CDI. Additionally, if patients required long-term care facility stay, like nursing homes or residential homes, episodes were also considered healthcare associated. Patients diagnosed at the hospital emergency services within 48 h without hospitalization, and no history of admission or discharge of a healthcare facility within the previous 12 weeks, were classified as CA-CDI, according to ECDC surveillance protocol [6].

Moreover, the 2018 IDSA/SHEA and ESCMID guidelines categorized case severity. An initial CDI episode is defined as a positive result obtained from culture, toxin, or molecular assay for a diarrheal stool with no other infection being diagnosed in the previous eight weeks [18,19]. Participants were classified as having either a non-severe (nsCDI), severe (sCDI) or recurrent (rCDI) episode, based on variables like laboratory analysis, clinical manifestations, disease severity and disease recurrence. Fulminant disease was not included in our analysis due to the absence of identified episodes. An initial nsCDI was defined as onset of symptoms with a positive diagnostic test and no history of infection within the previous eight weeks without any associated complications. On the other hand, an initial sCDI was classified based on laboratorial criteria, including the presence of marked leukocytosis, White Blood Cell (WBC) count above 15,000 cells/μL, elevated serum creatinine level (over 1.5 mg/dL), high blood lactate level, hypokalemia, and presence of fever (body temperature > 38.5 °C). Recurrent episodes were defined as a positive laboratory assay obtained between two and eight weeks after the diagnosis of a previous episode, with symptoms recurrence [33].

The response to antibiotic treatment schemes was assessed based on the resolution of diarrhea symptoms, with no further requirement for CDI treatment, improved analytical parameters, absence of new disease signs, and negative stool Toxin A and B samples, all the latter indicating CD eradication. Sustained cure was defined as treatment response without recurrence during follow-up period. Relapse or reinfection occurred when patients did not respond to first-line antibiotic therapy, with symptoms persistence and positive CD toxin assay within an 8-week period. We also aimed to compare annual incidence, mortality, and recurrence rates data obtained on this study with cohorts from different populations and time periods, documented in literature.

### 2.5. Statistical Analysis

We aim to compare and establish correlations between sociodemographic data, risk factors, comorbidities, and the incidence, severity and characteristics of disease. Episodes were categorized based on age groups, sex traits, antibacterial drug consumption, other risk factors and underlying diseases of interest, relating to its severity and recurrence rate. The overall incidence rate was determined as the number of patients with positive CD toxin assay per 10,000 patients admitted to our institution, using data obtained from the laboratory services (microbiology department), divided by the number of hospitalized patients, throughout 2018. All accessed from the hospital administrative management system.

The normal quantitative data were expressed as the mean ± standard deviation, median, minimum and maximum values. Descriptive statistics procedures were analyzed using Microsoft Excel^®^ 2013. Additionally, we used IBM^®^ SPSS V.25 statistics software to perform non-parametric correlation tests and multiple Spearman rank-order correlation tests to determine if there were any correlation between the different variables in our study.

### 2.6. Ethics

The study protocol was approved by the Ethics Committee of the Hospital do Espírito Santo de Évora (Approval ID number 042/22 of 3 May 2022) and data extraction was according to the Portuguese General Data Protection Regulation [34].

## 3. Results

### 3.1. Incidence and Disease Burden

During our study, a total of 395 patients were recorded with active diarrhea, suspected and therefore tested for CDI. Among them, 30 patients (7.6%) met the eligibility criteria and were included for further analysis, representing 0.2% of all hospitalizations. Also, a total of 14,493 patients were admitted to our hospital facility, resulting in an annual incidence rate of 20.7 cases per 10,000 hospitalized patients. 

### 3.2. Sociodemographic/Incidence and Disease Burden

Among the studied population, most episodes were healthcare-acquired, representing 13.1 cases per 10,000 admissions rate. The population was mostly composed of elderly patients. The average participants’ age was 76.4 ± 12.9 years [minimum 38–maximum 92 years], of which 83.3% were over the age of 65. There was a higher rate of female patients compared to male patients (rate ratio = 1.15), 53.3% and 46.7%, respectively. Table 1 illustrates the sociodemographic characteristics of patients, the site of disease acquisition, as well as the methods used for detection.

Among all patients, 63.3% (*n* = 19) were diagnosed during hospitalization or acquired CDI in another healthcare facility or long-term care unit and therefore classified as HA-CDI. In contrast, 36.7% (*n* = 11) of episodes were community-acquired. The majority of CDI cases were identified and diagnosed through the detection of CD Toxins A and B in stool samples, while only 6.7% were diagnosed through histological examination during colonoscopy. 

### 3.3. Clinical Outcomes and Risk Factors for CDI Development

Among the analyzed population, all participants presented risk factors that predisposed them to CDI and comorbidities that increased the risk for severe outcomes. Regarding the patients’ sociodemographic traits, Table 2 provides a summary of their underlying diseases and risk factors.

All patients in our study presented underlying conditions and comorbidities, resulting in polypharmacy, with six or more concomitant medications for the treatment of chronic illnesses and other medical conditions. The majority of participants, 96.7%, presented additional clinical conditions and multiple comorbidities, with cardiovascular diseases being the most predominant, especially arterial hypertension. Neoplastic diseases were also recorded in the studied population, as two patients presented metastatic colorectal cancer, both undergoing chemotherapy, and one patient receiving radiotherapy and chemotherapy for a neurological cancer (glioblastoma). Another participant had been diagnosed with non-Hodgkin lymphoma one year prior to CDI and was not currently undergoing chemotherapy. All participants presented multiple risk factors. Among the analyzed risk factors, the most important and prevalent was high-risk antibiotic consumption in the 12-week period prior to infection. Furthermore, continuous PPIs therapy within eight weeks before or during the observation period was recorded in most of patients. 

### 3.4. Infection Characteristics

According to IDSA/SHEA guidelines case definitions, episodes were classified as HA-CDI or CA-CDI based on their origin. However, the management of CDI is commonly discussed in terms of severity and number of episodes. Therefore, for infection severity, we assessed analytical, clinical, and physiological criteria, resulting in the classification of initial non-severe and severe CDI episodes, or recurrences. Regarding infection severity, Table 3 provides a summary of its distribution and characterization.

Among all patients, 60.0% met eligibility criteria for nsCDI, presenting symptom onset with a positive diagnostic test, no recent infection within the previous eight weeks, and absence of severe analytical characteristics or symptoms. Severe episodes, characterized by the presence of fever, marked leukocytosis, rise in creatinine and high lactate serum level at presentation, were observed in 23.3% of participants. Recurrent infections, characterized as a recurrence of symptoms within eight weeks or less of a primary episode, were recorded in 16.7% of cases. All recurrences were healthcare associated, and most community-acquired cases were characterized as non-severe episodes. Additionally, we also conducted an analysis considering demographic factors such as age group, sex, comorbidities, number of risk factors, and annual distribution, as summarized in Table 4. 

Throughout our research, most patients presented fever, and elevated analytical parameters, including increased WBC count and above-normal serum lactate levels. High serum creatinine levels and hypokalemia was also presented in most sCDI cases, requiring serum electrolyte monitoring and correction. These analytical markers were more consistently observed in sCDI and rCDI episodes, all exhibiting fever and marked leukocytosis at presentation. There were no significant differences in the overall CDI distribution for our participants, however recurrent CDI was more prevalent in female subjects. Additionally, HA-CDI was also more frequently recorded in females, while CA-CDI was more frequent in male patients. Most of the studied participants were aged over 65 years. These patients exhibited more marked electrolyte imbalance, requiring serum monitoring and correction, with 71.4% of sCDI cases developing hypokalemia. Severe episodes were mostly reported in elderly patients (85.8% of participants). Consistently, recurrent episodes were also more prevalent in older patients, since 80.0% of these episodes were recorded in patients over 80 years old. Considering the site of acquisition, healthcare-related infections were more frequent in advanced age subjects, when compared with CA-CDI.

We recorded the number and types of comorbidities and underlying conditions of each patient, and then characterized them using the Charlson Comorbidity Index (CCI) classification. All studied subjects presented multiple underlying conditions of interest, with 76.7% of participants documenting three or more comorbidities. The median CCI score was 9.1 ± 1.6 for all participants (ranging from 5 to 15), with 63.3% of patients scoring 9.0 or higher. Among patients aged above 65 years, a higher CCI score was identified when compared to younger patients. Furthermore, patients that developed severe episodes presented higher CCI scores, in the range of 10–12, when compared with nsCDI and rCDI cases. 

A Multiple Spearman’s correlation test was run to assess the relationship between the Charlson Classification Index and infection criteria for CDI from our sample, shown in Table 4. There were statistically significant correlations between serum lactate levels and CCI *p* = 0.023.

Risk factors were evident among all study participants, irrespective of their severity and origin of CDI, with prior antibiotic exposure emerging as the predominant factor. Most participants were exposed to multiple antibiotic drugs during the study. All sCDI and rCDI episodes had previous antimicrobial consumption. Moreover, only one patient with a non-severe episode and a CA-CDI had no record of previous antibiotic consumption. Exposure to multiple antibiotic drugs were also more prevalent in HA-CDI cases, when compared to CA-CDI episodes. Additionally, all severe episodes were developed by subjects who were previously exposed to antibiotics and aged over 65 years old, simultaneously. The presence of multiple risk factors was recorded in 86.7% of all patients at presentation, accounting for all sCDI recorded episodes and 80.0% of recurrences. Also, patients hospitalized due to CDI spent on average 26 days at our hospital.

CDI incidence was evenly distributed across all the period of study, but more prevalent in the first trimester of the year. Severe episodes and recurrences were mostly recorded in the second and third trimester of the year, compared to nsCDI, which were mostly healthcare-acquired. This research revealed a mortality rate of approximately 23.3%, mostly in older participants, especially those aged over 65 years. The latter occurred in participants meeting multiple underlying diseases and risk factors at presentation. It was also more frequent in healthcare-acquired episodes. Finally, all deaths occurred in subjects pre-exposed to high-risk antibiotic drugs.

### 3.5. Medication Profile

Cumulative antibiotic exposure, use of multiple antibacterial agents, and increased therapy days all contributed to increased risk of CDI development. Antibiotics such as clindamycin, third-generation cephalosporin, amoxicillin, and fluoroquinolones have been described to be related to higher infection risks [8]. Our study revealed that most patients had prior antibiotic consumption, with previous history of treatment with multiple antibacterial agents identified (mean: 2.03 ± 0.72). According to the World Health Organization, the consumption of antibacterials is expressed by the Anatomical Therapeutic Chemical (ATC) classification system classes J01 group, which comprises all antibacterials for systemic use. Table 5 summarizes overall antibiotic exposure in the 12-week period prior to CDI, and distribution based on infection severity or origin.

According to the 2018 report from INFARMED—National Authority for Medicament and Health Products, I.P., during this study, our hospital institution recorded an antibiotic consumption index corresponding to 4.4% of all drugs consumption. This represents a total of 150,222 units, accounting for 35.8% of overall antimicrobial consumption in the region of our study, and 1.5% of all hospital-associated antibiotic consumption in Portugal in 2018 [35]. Their report also provided information on antibiotic exposure in our hospital facility based on ATC classification. Additionally, this report documented a cephalosporin consumption index of 21.5%, similar to the exposure rate presented in this work. Our patients also exhibited higher rates of fluoroquinolones and carbapenems consumption, with rates of 43.3% and 20.0%, respectively. 

We aim to assess antibiotic consumption in our sample and analyze the prevalence according to the origin and severity of CDI episodes. All patients who developed sCDI or rCDI received antibacterial therapy in the 12-week period prior to infection. All community-acquired CDI cases were also associated with antibiotic use, with penicillin derivatives being the most common. Furthermore, 26.7% of all participants had concomitant treatment with penicillin derivatives and fluoroquinolones, while 13.3% were treated with both penicillin and cephalosporin, all the latter high-risk antibiotics. Previous exposure with clindamycin (one of the higher-risk antibiotics) was exclusively identified in patients with sCDI and rCDI, both healthcare-associated. 

### 3.6. Pharmacological Treatment Effectiveness

The main goal of treatment is resolution of diarrhea and prevention of recurrence, thus decreasing the burden of disease. The initial step in managing CD eradication involves minimizing unnecessary antibiotic exposure and immediate discontinuation of inciting antimicrobial agents. Treatment regimens may vary depending on severity of the episode at presentation and number of recurrences. Sustained cure or effectiveness of antibiotic therapy was defined as the absence of symptoms, recurrence or reinfection during the 8-week follow-up period, with a negative stool Toxin A and B assay. We analyzed the overall success of treatment protocols as a first-line antibiotic therapy and specifically for each severity and origin classification of the disease. Thus, Table 6 provides a summary of the management of CDI first-line treatment based on severity and origin and its corresponding effectiveness.

Successful outcomes were achieved with patients treated with metronidazole (first line of treatment for most cases) as an isolated antimicrobial agent, indicated by a sustained cure in most cases and high effectiveness in CD eradication, regardless of infection severity. Mainly recurrent cases were refractory to metronidazole in monotherapy. In contrast, oral vancomycin was employed as a single antibiotic agent to treat fewer CDI episodes within our studied population. Additionally, CD demonstrated resistance to vancomycin regimens, leading to treatment failure and subsequent disease relapse, when used as a first-line agent for both nsCDI and rCDI episodes. Also, all patients treated with vancomycin were healthcare-associated. Finally, the combination treatment scheme of metronidazole and vancomycin was consistently effective regardless of case severity or origin, especially in sCDI and HA-CDI episodes. Other supportive measures, such as rehydration and correction of electrolyte imbalances were implemented for all patients diagnosed with CDI, despite severity classification or type of treatment.

### 3.7. The Role of Probiotics in CDI

Our research showed that most episodes (63.3%) were treated with combined SoC antibiotic regimens and probiotics, documented mostly in female participants. Among these cases, 79.9% achieved CD eradication and did not experience refractory or recurrence of disease. Most sCDI cases (71.4%) and all rCDI episodes received probiotic therapy either to prevent further recurrence events or to enhance treatment effectiveness, when compared to primary nsCDI episodes. Notably, all episodes treated with probiotics showed higher effectiveness rates in CD eradication, regardless of the severity of the infection.

### 3.8. Correlation Analysis 

We took information from Table 3, Table 4 and Table 5 and other collected information and ran a multiple correlation matrix. Variables considered for this test were sex, age, Charlson Index, antibiotic intake, CDI site of acquisition and severity, refractory therapy, initial antibiotic treatment and probiotics use. Our Multiple Spearman’s correlation indicated, from all the thirty patients, there were statistically significant correlations in CDI Severity vs. Use of Probiotics (*p* = 0.003), Refractory to Initial Therapy vs. Use of Probiotics (*p* = 0.003), Sex vs. Use of Probiotics (*p* = 0.05) and Site of Acquired Infection vs. Sex (*p* = 0.002).

### 3.9. Epidemiological Tendency of CDI Episodes at HESE (2018–2022)

The focus of this study was to evaluate the epidemiology of CDI and assess the clinical impact of this event on hospitalized patients during 2018. Additionally, we also conducted an analysis focusing only on infection incidence during an extended four-year period, using patients’ sociodemographic data, meeting the same eligibility criteria and case origin information. This additional descriptive review performed between 2019 and 2022, detailing CDI incidence, and both healthcare and community-acquired prevalence, in addition with data from another study performed in our hospital institution, by Silva et al. 2012, is summarized on Table 7 [36]. The purpose of this extended period of analysis was to confirm the increasing CDI incidence tendency reported in the literature, particularly in CA-CDI cases. 

Over this four-year period, a total of 58,099 patients were admitted and hospitalized at our hospital facility, and we recorded 134 positive cases of CDI, accounting for an average annual incidence rate of 23.1 cases per 10,000 hospitalizations (ranging from 18.1 to 28.5 cases per 10,000 patients). A consistent pattern in elderly participants was recorded, with an average age of 77.0 ± 9.9 years (87.3% over the age of 65), consistent with the 2018 data. Silva et al. (2012), reported an increase in CDI incidence at the same hospital facility, during the early 2000’s. An 8-fold increase in incidence was documented during an eight-year period (2000 to 2008), recording 16.0 cases per 10,000 hospitalized patients in 2008, most of them being healthcare-acquired [36]. Throughout our investigation, an increase in community-acquired infections prevalence compared to healthcare-acquired was observed. Figure 1 shows a visual representation of the evolution of overall CDI incidence, during a twenty two-year period, and both HA-CDI and CA-CDI prevalence at our hospital institution, since 2007.

As documented by Silva et al., (2012) [36] and our study, overall CDI episodes incidence consistently increased in our institution overtime, as observed with the prevalence of CA-CDI episodes. These findings align with other cohorts indicating an increasing trend in community-acquired episodes, with a 2.5-fold increase since 2007, and within our period of study, accounted for 55.9% of all CDI cases, in 2022.

### 3.10. Comparative Epidemiology Analysis

Epidemiological data regarding CDI in Portugal remains limited; therefore, we conducted a literature review, focusing on epidemiological details. Eleven studies were included based on our predefined inclusion criteria for this descriptive review. Table 8 summarizes disease overall incidence rate (number of CDI cases per 10,000 admissions) and antibiotic consumption rate (AMC percentage, %) in this study as well as in Portugal, Europe, and North America.

We compare the results documented on this research with published data, which pointed to a sustained increase in overall CDI incidence rate, especially community-acquired episodes and a decrease in healthcare-acquired cases over the past decade [3,7]. Furthermore, we compared antibiotic intake prior to infection.

## 4. Discussion

### 4.1. Infection Distribution by Severity and Origin 

Regarding case severity and CDI epidemiological origin, our results revealed that healthcare-associated infections were more aggressive, as all recurrent episodes and the majority of severe cases (57.1%) were described as HA-CDI, also supporting the evidence from other studies [47]. Recurrent episodes were consistently lower in community-associated infections, compared to healthcare-associated, as per the literature [3,5,7]. In contrast, most community-associated CDI cases exhibited more indolent episodes, with 72.7% being classified as nsCDI, and no recurrence being associated. Patients diagnosed with CA-CDI were younger and reported less antibiotic exposure during the 12 weeks prior to diagnosis, as shown by Khanna et al. (2012) [48]. 

### 4.2. Risk Factor and CDI Severity 

During the observation period all patients had known risk factors for CDI development, with antibiotic exposure as the most important and modifiable factor. Our study revealed the highest CDI association with antibiotic consumption rate, compared to other Portuguese cohorts. All patients who developed severe episodes and recurrences, representing all CA-CDI cases, had pre-exposure to antibiotic drugs in the 12-week period preceding the infection. Our studied population had a fluoroquinolones and carbapenems consumption 3.5 times higher than the reported by the Portuguese National Authority of Medicines for our institution and period of study [35]. Notably, cephalosporin consumption was most commonly associated with severe episodes. Furthermore, all clindamycin-associated infections were healthcare-acquired, and these patients experienced more severe infections. All patients pre-treated with clindamycin experienced treatment failure and disease relapse, irrespective of the drug used for CD eradication. 

Studies have demonstrated that specific strains of probiotics can reduce the risk, severity, and duration of diarrhea in CDI patients as well as decrease the recurrence rate of the infection [26]. In our study, 63.3% of patients received probiotics in conjunction with SoC antibiotics. Remarkably, these patients demonstrated a high cure rate, with no relapse of infection being recorded, particularly among female subjects. While its effectiveness in preventing and treating CDI remains limited, our findings establish a strong correlation between CDI severity and probiotic use. Therefore, our study contributes to the existing but limited literature, supporting the potential benefits of probiotics in managing CDI. Therefore, physicians should be more sensitive and alert in prescribing both prebiotics and probiotics during antibiotic therapy. This finding may explain the higher proportion of females receiving probiotics compared to males in our study and the correlation observed between gender and probiotic usage, as documented by Liu et al., 2020 and Yoon & Kim, 2021 [25,49]. 

Older patients are more susceptible to CDI compared to younger patients both in terms of incidence and mortality, due to changes in microbial gut diversity, weakened immune systems, presence of comorbidities, and overall frailty [49]. Our findings reinforce that susceptibility, as most of the included patients were aged over 65 years. This highlights the higher prevalence of severe cases and recurrences in the elderly population, both occurring mostly in individuals over 80 years old. This investigation revealed an overall mortality rate of 23.3%, mainly observed in patients with multiple underlying diseases and risk factors, as supported by the literature. This event was consistently observed in patients with previous antibiotic exposure. Regarding the seasonal distribution, HA-CDI episodes were more predominant in the second and third trimesters of the year. This pattern may suggest a possible nosocomial outbreak during this period, as more aggressive CDI episodes and recurrences were documented during these two trimesters, within our study period.

Studies suggest that CDI affects both men and women equally. Within our sample, there was no significant difference in CDI distribution overall, regardless of the participants’ gender. Esteban-Vasallo et al. (2016) and more recently, Younas et al. (2021), analyzed the association between gender and CDI acquisition, suggesting a higher association of CA-CDI with women. However, our findings indicate a stronger statistical correlation between female patients and HA-CDI, which can partially explain the higher recurrence rates observed in women, associated with more resistant CD strains [50,51]. 

Community-associated CDI was more prevalent in male patients. Moreover, recurrent CDI episodes were consistently more prevalent in females and severe episodes were more frequent in males. These findings are in agreement with other studies, indicating that women are more susceptible to recurrence, due to variations in microbiota composition influenced by hormonal factors and differences in immune response mechanisms, that may increase the risk of CD colonization and overgrowth [52,53]. 

Regarding the correlation between CCI and hyperlactatemia, lactate levels have long been recognized as a crucial factor linked to poor prognosis in critically ill patients. While hyperlactatemia upon ICU admission has been established as a reliable prognostic marker, subsequent changes in lactate concentration have also been proven to possess independent predictive value. The initial lactate level serves as a valuable target in patients suspected of infection or septic shock, offering insights into their morbidity and mortality risk within a 30-day period. Therefore, the effect of serum lactate levels could influence in greater proportion the associated comorbidities and outcome of our patient sample [54,55].

### 4.3. Pharmacological Treatment Effectiveness

Our study documented that physicians, regardless of case severity, prescribed metronidazole as first-line treatment, following the 2018 guidelines. However, multiple studies have reported a decreased effectiveness in eradicating CD, updating those guidelines, in which metronidazole is no longer recommended as a first-line agent for CDI treatment [18,19]. Our study summarized real-world data regarding treatment schemes effectiveness within our study population. Interestingly, our findings demonstrated that metronidazole was highly effective in eradicating CD in most initial non-severe and severe cases, supporting the 2018 guidelines. All CA-CDI episodes were also successfully treated with metronidazole. Conversely, all patients treated with oral vancomycin experienced treatment failure and refractory CDI, regardless of the severity of the infection. All of these infections were healthcare-associated, which could explain the high refractory rate and vancomycin treatment failure. As described in Table 3, these patients developed most of the severe infections and all recurrences, probably related to patient-to-patient transmission of certain resistant strains to SoC antibiotics. Also, these patients were previously treated for CDI, which could result in antibiotic resistance. Our findings documented the efficacy of combined double treatment regimens, using both metronidazole and vancomycin, which proved highly effective in CD eradication. No relapses were recorded during the period of study, when this dual antibiotic scheme was used as either initial treatment or after treatment failure.

### 4.4. Overall Infection Distribution by Country and Study Period

The overall incidence rate of CDI has shown a substantial increase over time. Our study showed a 10.4-fold increased incidence of this event over an eighteen-year period [36]. Similar findings were reported in studies conducted during the early 2000s, where CDI incidence rates were lower worldwide but displayed a rising trend until 2007 [5]. Additionally, identical incidence rates have been documented in cohorts worldwide during a time period close to our study, indicating an increase in infection rates compared to previous decades [3,7,55]. 

A study involving six Portuguese public hospital centers, from 2017, found an incidence rate of 20.2 per 10,000 admissions, consistent with our findings [39]. Another study conducted at the North Lisbon University Hospital Center, found an incidence rate of 15.4 cases per 10,000 patients in 2007, similarly to the previous report from our hospital in 2008 [36,43]. Additionally, the EUCLID study group published a multi-center, prospective bi-annual point prevalence study of CDI in hospitalized patients, revealing an average incidence rate of 14.7 cases per 10,000 patients in Portuguese participating centers, between 2011 and 2013, supporting the epidemiological tendency of the disease [56]. 

Our additional analysis conducted between 2019 and 2022 documented higher incidence rates than in previous years. The highest incidence rate observed at *HESE* was recorded in 2019, with 28.5 cases per 10,000 hospitalizations documented, representing a 37.7% increase compared to the 2018 study period and a 14.3-fold increase from the 2000 data [36]. The latter represents the highest incidence ever recorded at our hospital, regardless of the period of study, confirming the increasing trend of this event. It was also the second-highest incidence rate reported in Portugal, according to Correia et al. (2012), recording 86.6 cases per 10,000 patients [57]. Other European studies, like the one from Gastmeier et al. (2009) in Germany, reported a higher CDI incidence rate, with 46.5 cases per 10,000 patients documented, 2.2 times higher than the findings in this study [45]. 

The antibiotic consumption index prior to infection in our investigation was consistent with findings from cohorts in other Portuguese hospitals. We observed an antibiotic consumption rate of 96.7% among participants in the 12 weeks preceding CDI infection. This agrees with ORCHID, a multicentric study involving 97 European hospitals, of which eight Portuguese Hospitals and collaborators, reported AMC rates ranging from 70.0% to 100.0%, with a mean of 92.0% [58].

### 4.5. Epidemiological Tendency of HA-CDI and CA-CDI Episodes at HESE

Our documented data suggests a significant increase in overall CDI incidence until 2019 followed by a considerable decrease thereafter. Throughout our investigation, and consistent with other research, a higher incidence of healthcare-acquired infections compared to community-acquired was observed during the last decades. This epidemiological pattern has changed and currently CA-CDI achieved a higher prevalence rate. Our study corroborates this pattern, since community cases consistently increased in our institution, while HA-CDI incidence remained relatively stable and decreased more recently. Treglia et al. have investigated the potential impact of influenced healthcare practices, policies, and infection control measures, which may offer partial explanation for the decline in HA-CDI incidence observed in our research [59]. Similarly, other Portuguese authors reported a rise in CA-CDI incidence, compared to previous years [37,39]. This event was also recorded in other studies worldwide. Additionally, a plateau in HA-CDI was documented between 2007 and 2018, with a subsequent decrease until 2022, also supported by the literature [3,60]. The ECDIS Study Group with the support of ECDC and members of a study group of ESCMID designed a study in 2008 recording epidemiological data from 34 European countries, indicating an annual CDI incidence rate of 13.0 per 10,000 admissions in Portugal, with 86.0% being healthcare-associated, in accordance with our findings [42].

### 4.6. Infection Burden during COVID-19 Pandemic

Data available from January 2019 to September 2021 indicate a lower annual incidence rate of CDI, particularly healthcare-acquired [61]. This decrease is probably related to improved adherence to Good Practices in Infection Prevention and Disease Control measures in healthcare settings applied during the COVID-19 pandemic. Such practices included enhanced hand hygiene, environmental cleaning, patient isolation, and increased use of personal protective equipment [61,62]. Our findings agree with the literature, as the overall incidence significantly decreased during the pandemic, since the lowest incidence rate was documented in 2021, with 18.1 cases per 10,000 hospitalized patients, indicating a 57.5% decrease compared to the incidence observed in 2019. Additionally, HA-CDI episodes showed the most marked decrease in prevalence, during this period, supporting this hypothesis. 

### 4.7. Study Limitations

This study had several limitations. First, it was a retrospective observational intervention with a small sample size, and the methodology relied on the review of clinical reports, which may introduce biases and incomplete data. The estimation of hospitalizations per year was based on rates, which may not accurately reflect the true number of cases. The assessment of CD acquisition origin was not always reliable, potentially affecting the accuracy of the findings. Study methodology might be responsible for a lower recurrence rate, since the follow-up of episodes was not optimal and the focus of this investigation was only placed in hospitalized patients, not capturing data after discharge or any admission to another healthcare facility. We assume a possible under or overestimation of cases, due to less specific methods of analysis used in previous years. The use of more sensitive CD assays, like Nucleic Acid Amplification Tests, introduced the potential for increased detection rates. Antimicrobial susceptibility testing is not routinely performed for CD and data evaluating minimum inhibitory concentrations are limited. By performing genome sequencing surveillance, the optimal antibiotic treatment scheme will be provided to patients, according to EUCAST database cut-off values for each antimicrobial agent. Another limitation was the inconsistency among studies in epidemiological definitions on incidence rates, which limited the comparative analysis. Furthermore, incidence data showed large variations across countries, which may be attributed to different reporting practices or different diagnostic tests used, rather than raw incidence rate differences. Moreover, the evaluation of therapeutic effectiveness relied on symptom resolution, without subsequent hospitalization and a negative laboratory assay for CD toxins in stools being recorded. Finally, treatment failure was determined based on recurrence, relapse, or reinfection within an eight-week period.

## 5. Conclusions

Our study provides a comprehensive description of the epidemiology of CDI and assesses the clinical impact at a Portuguese hospital. Our investigation revealed a 10.4-fold increase compared with early 2000’s data from our hospital facility, agreeing with previous reports, highlighting the exponential increase in infection incidence both in Portugal and worldwide. This increase was particularly notable in community-acquired episodes, requiring further investigation focusing on this incidence pattern. Moreover, the antibiotic consumption index, mortality, and recurrence rates identified in our study were also consistent with the literature. These findings underscore the urgent need for antimicrobial stewardship programs. The involvement of clinical pharmacists in these intervention measures is crucial for the judicious use of antibacterial agents, that focus on antibiotic restriction ensuring appropriate use, avoiding medication error and following guidelines to prevent prolonged antibiotic treatment after bacterial elimination.

The adherence to best practices for environmental cleaning in healthcare facilities and infection prevention and control measures, have played a crucial role in preventing CD transmission, mostly in healthcare-acquired infections. By adhering to the most up-to-date guidelines for CDI management, utilizing the most cost-effective treatment strategies available, and prioritizing the sustainability of healthcare systems, a positive impact on disease management and control will be achieved. Further analysis will be conducted at our hospital institution after the implementation of the most updated CDI treatment and prevention strategies, focusing on the cost-effectiveness ratio and security profile. Finally, the potentiality of our research could imply future studies on effectiveness of probiotics in CDI treatment, incidence and prevalence of CA-CDI, newer CDI diagnostic methods, to name a few.

## Figures and Tables

**Figure 1 healthcare-12-00076-f001:**
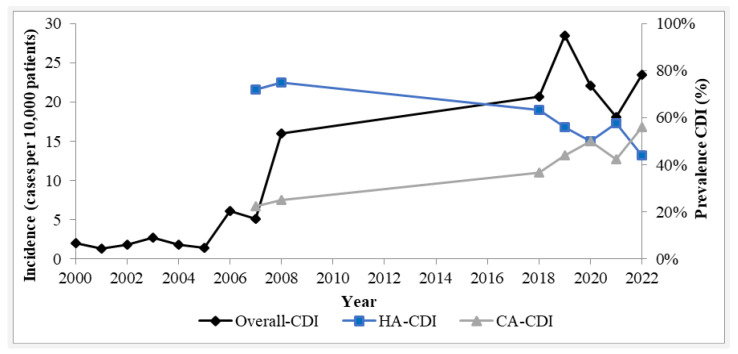
CDI episodes distribution in hospitalized patients at HESE (2000–2022).

**Table 1 healthcare-12-00076-t001:** Patient’s sociodemographic characteristics, infection acquisition site and detection methods.

Total Sample Size *n* = 30		No. (%)
Sex of participants	Female	16 (53.3)
	Male	14 (46.7)
Group age of participants	<65 years old	5 (16.7)
	65–80 years old	11 (36.7)
	≥80 years old	14 (46.7)
Site of Acquisition of CD	Healthcare-acquired CDI	19 (63.3)
	Community-acquired CDI	11 (36.7)
Detection Methods	Detection of CD Toxins A and B	28 (93.3)
	Histological examination	2 (6.7)

**Table 2 healthcare-12-00076-t002:** Baseline patient risk factors and comorbidities.

Total Sample Size *n* = 30		No. (%)
Patients’ risk factors	Antibiotic therapy < 12 weeks	29 (96.7)
	Age > 65 years old	25 (83.3)
	Hospitalization < 8 weeks	19 (63.3)
Patients’ concomitant medication	Any antibiotic consumption	29 (96.7)
	Proton-pump inhibitors or Gastric acid suppressants	18 (60.0)
	Immunosuppressant drugs	6 (20.0)
	Continuous NSAIDs	3 (10.0)
Underlying conditions/ Comorbidities	≥1 Comorbidity of interest	29 (96.7)
	Arterial hypertension	26 (86.7)
	Surgery < 2 months	15 (50.0)
	Obesity	5 (16.7)
	Peptic gastric ulcer	6 (20.0)
	Nasogastric Tube	5 (16.7)
	Diabetes	14 (46.7)
	Immunocompromised	6 (20.0)
	Dementia	20 (66.7)
	Cancer/Neoplastic diseases	4 (13.3)
	HIV/AIDS	1 (3.3)
	Moderate to severe hepatic impairment	12 (40.0)

**Table 3 healthcare-12-00076-t003:** Distribution of CDI cases by its origin and severity classification.

*C. difficile* Distribution		*C. difficile* Origin Classification	
*C. difficile* Severity Classification	Overall CDI *n* = 30, No. (%)	HA-CDI, *n* = 19, No. (%)	CA-CDI, *n* = 11, No. (%)
Non-Severe CDI (nsCDI)	18 (60.0)	10 (52.6)	8 (72.7)
Severe CDI (sCDI)	7 (23.3)	4 (21.1)	3 (27.3)
Recurrent CDI (rCDI)	5 (16.7)	5 (26.3)	0 (0.0)

**Table 4 healthcare-12-00076-t004:** Patients’ data distributed by CDI severity and origin classification.

*C. difficile* Classification (Sample Size)		Overall CDI (*n* = 30)	nsCDI (*n* = 18)	sCDI (*n* = 7)	rCDI (*n* = 5)	HA-CDI (*n* = 19)	CA-CDI (*n* = 11)
Number of Patients, (%)		No. (%)	No. (%)	No. (%)	No. (%)	No. (%)	No. (%)
Analytical criteria	Fever (>38.5 °C)	21 (70.0)	9 (50.0)	7 (100.0)	5 (100.0)	14 (73.7)	7 (63.6)
	WBC > 15,000	20 (66.7)	9 (50.0)	7 (100.0)	5 (100.0)	13 (68.4)	7 (63.6)
	High creatinine	15 (50.0)	8 (44.4)	6 (85.7)	1 (20.0)	9 (47.4)	6 (54.5)
	Hypokalemia	20 (66.7)	12 (66.7)	5 (71.4)	3 (60.0)	13 (68.4)	7 (63.6)
	Hyperlactatemia	14 (46.7)	10 (55.6)	3 (42.9)	1 (20.0)	8 (42.1)	6 (54.5)
Sex of participants	Female	16 (53.3)	9 (50.0)	3 (42.9)	4 (80.0)	14 (73.7)	2 (18.2)
	Male	14 (46.7)	9 (50.0)	4 (57.1)	1 (20.0)	5 (26.3)	9 (81.8)
Group age of participants	30–48 years	2 (6.7)	1 (5.6)	0 (0.0)	1 (20.0)	2 (10.5)	0 (0.0)
	48–64 years	3 (10.0)	2 (11.1)	1 (14.3)	0 (0.0)	0 (0.0)	2 (18.2)
	65–80 years	11 (36.7)	8 (44.4)	3 (42.9)	0 (0.0)	7 (36.8)	5 (45.5)
	>80 years	14 (46.7)	7 (38.9)	3 (42.9)	4 (80.0)	10 (52.6)	4 (36.4)
Charlson Comorbidity Index	CCI 5–6 Score	3 (10.0)	2 (11.1)	1 (14.3)	0 (0.0)	1 (5.3)	2 (18.2)
	CCI 7–8 Score	8 (26.7)	5 (27.8)	2 (28.6)	1 (20.0)	5 (26.3)	3 (27.3)
	CCI 9–10 Score	15 (50.0)	9 (50.0)	2 (28.6)	4 (80.0)	10 (52.6)	5 (45.5)
	CCI 11–12 Score	2 (6.7)	0 (0.0)	2 (28.6)	0 (0.0)	2 (10.5)	0 (0.0)
	CCI > 12 Score	2 (6.7)	2 (11.1)	0 (0.0)	0 (0.0)	1 (5.3)	1 (9.1)
Number of risk factors	One	4 (13.3)	3 (16.7)	0 (0.0)	1 (20.0)	3 (15.9)	1 (9.1)
	Two	12 (40.0)	6 (33.3)	4 (57.1)	2 (40.0)	6 (31.6)	6 (54.5)
	Three	10 (33.3)	6 (33.3)	3 (42.9)	1 (20.0)	7 (36.8)	3 (27.3)
	≥Four	4 (13.3)	3 (16.7)	0 (0.0)	1 (20.0)	3 (15.9)	1 (9.1)
Antimicrobial Consumption (AMC)	None AMC	1 (3.3)	1 (5.6)	0 (0.0)	0 (0.0)	0 (0.0)	1 (9.1)
	One AMC	8 (26.7)	4 (22.2)	3 (42.9)	1 (20.0)	4 (21.1)	4 (36.4)
	Two AMC	16 (53.3)	11 (61.1)	2 (28.6)	2 (40.0)	10 (52.6)	5 (45.5)
	≥Three AMC	5 (16.7)	2 (11.1)	2 (28.6)	2 (40.0)	5 (26.3)	1 (9.1)
CDI year distribution	1st trimester	9 (30.0)	7 (38.9)	1 (14.3)	1 (20.0)	3 (15.8)	6 (54.5)
	2nd trimester	8 (26.7)	2 (11.1)	4 (57.1)	2 (40.0)	6 (31.5)	2 (18.2)
	3rd trimester	8 (26.7)	4 (22.2)	2 (28.6)	2 (40.0)	6 (31.5)	2 (18.2)
	4th trimester	5 (16.7)	5 (27.8)	0 (0.0)	0 (0.0)	4 (21.1)	1 (9.1)
Mortality rate	Clinical endpoint	7 (23.3)	2 (11.1)	4 (57.1)	1 (20.0)	5 (26.3)	2 (18.2)

**Table 5 healthcare-12-00076-t005:** Antibiotic consumption distributed by CDI classification.

ATC Classification	Antibiotic Classes	Overall, No. (%)	nsCDI, No. (%)	sCDI, No. (%)	rCDI, No. (%)	HA-CDI, No. (%)	CA-CDI, No. (%)
J01	Any antibiotic	29 (96.7)	17 (94.4)	7 (100.0)	5 (100.0)	18 (94.7)	11 (100.0)
J01C/J01CR	Penicillin derivatives	18 (60.0)	11 (61.1)	5 (71.4)	2 (40.0)	11 (57.9)	7 (63.6)
J01M	Fluoroquinolones	13 (43.3)	8 (44.4)	3 (42.9)	2 (40.0)	8 (42.1)	5 (45.5)
J01D	Cephalosporins	8 (26.7)	7 (38.9)	0 (0.0)	1 (20.0)	6 (31.6)	2 (18.2)
J01DH	Carbapenems	6 (20.0)	4 (22.2)	2 (28.6)	0 (0.0)	3 (15.9)	3 (27.3)
J01G	Aminoglycosides	4 (13.3)	2 (11.1)	2 (28.6)	0 (0.0)	3 (15.9)	1 (9.1)
J01FF	Clindamycin	2 (6.7)	0 (0.0)	1 (14.3)	1 (20.0)	2 (10.5)	0 (0.0)
J01FA	Macrolides	1 (3.3)	0 (0.0)	0 (0.0)	1 (20.0)	1 (5.3)	0 (0.0)
J01B, J01G, J01X	Other antibiotic	9 (30.0)	2 (11.1)	2 (28.6)	5 (100.0)	7 (36.8)	2 (18.2)

**Table 6 healthcare-12-00076-t006:** Management of CDI treatment schemes and its effectiveness.

Antibiotic Drug (ATC Classification)	Metronidazole J01XD01		Vancomycin J01XA01		Metronidazole/Vancomycin J01XD01/J01XA01	
*C. difficile* Classification	First-Line No. (%)	Effectiveness No. (%)	First-Line No. (%)	Effectiveness No. (%)	First-Line No. (%)	Effectiveness No. (%)
Overall CDI (*n* = 30)	26 (86.7)	16 (61.5)	3 (10.0)	0 (0.0)	1 (3.3)	1 (100.0)
nsCDI (*n* = 18)	16 (88.9)	11 (68.8)	2 (11.1)	0 (0.0)	0 (0.0)	N/A
sCDI (*n* = 7)	6 (85.7)	4 (66.7)	0 (0.0)	N/A	1 (14.3)	1 (100.0)
rCDI (*n* = 5)	4 (80.0)	1 (20.0)	1 (20.0)	0 (0.0)	0 (0.0)	0 (0.0)
HA-CDI (*n* = 19)	15 (78.9)	10 (66.7)	3 (15.8)	0 (0.0)	1 (5.3)	1 (100.0)
CA-CDI (*n* = 11)	11 (100.0)	6 (54.4)	0 (0.0)	N/A	0 (0.0)	N/A

**Table 7 healthcare-12-00076-t007:** CDI episodes distribution in hospitalized patients at HESE (2000–2022).

Author’s Name/ Study Period	Overall CDI Incidence	HA-CDI, No. (%)	CA-CDI, No. (%)
Present study, 2022	23.5 cases per 10,000	10.4 (44.1)	13.1 (55.9)
Present study, 2021	18.1 cases per 10,000	10.4 (57.7)	7.7 (42.3)
Present study, 2020	22.2 cases per 10,000	11.1 (50.0)	11.1 (50.0)
Present study, 2019	28.5 cases per 10,000	16.0 (56.1)	12.5 (43.9)
Present study, 2018	20.7 cases per 10,000	13.1 (63.3)	7.6 (36.7)
Silva et al., 2012 (2008) [36]	16.0 cases per 10,000	12.0 (75.0)	4.0 (25.0)
Silva et al., 2012 (2001–2007) [36]	5.2 cases per 10,000	16.0 (77.2)	5.0 (22.8)
Silva et al., 2012 (2000) [36]	2.0 cases per 10,000	N/A	N/A

**Table 8 healthcare-12-00076-t008:** Annual overall incidence of CDI and antibiotic consumption rates in other cohorts.

Author’s Name/ Reference	Study Period	Research Location	Overall CDI Incidence	AMC (%)
Present Study (4-year period)	2019–2022	HESE	23.1/10,000 patients	N/A
Present Study	2018	HESE	20.7/10,000 patients	96.7
Barbosa-Martins et al. [37]	2018	HSO	4.8/10,000 patients	68.4
Teixeira et al. [38]	2017	Portugal	9.0/10,000 patients	N/A
Nazareth et al. [39]	2017	Portugal	20.2/10,000 patients	86.0
Balsells et al. [40]	2016	United States	22.4/10,000 patients	N/A
Sintra et al. [41]	2010	CHUC	21.6/10,000 patients	95.8
Silva et al. [36]	2008	HESE	16.0/10,000 patients	91.2
Bauer et al. [42]	2008	Portugal	13.0/10,000 patients	79.0
Vieira et al. [43]	2007	CHLO	15.4/10,000 patients	82.0
Asensio et al. [44]	2007	Spain	12.2/10,000 patients	40.7
Gastmeier et al. [45]	2007	Germany	46.5/10,000 patients	60.0
Cardoso et al. [46]	2004	HFF	4.3/10,000 patients	71.0
Silva et al. [36]	2000	HESE	2.0/10,000 patients	91.2

Abbreviation: AMC: antimicrobial consumption; HESE: Hospital do Espírito Santo, Évora; HSO: Senhora da Oliveira Hospital—Guimarães; CHUC: Coimbra University Hospital; CHLO: Western Lisbon Hospital Center; HFF: Hospital Professor Doutor Fernando Fonseca, Amadora; N/A: Not applicable.

## Data Availability

The datasets generated and/or analyzed during the current study are not publicly available due to protecting the privacy of study participants, according to the Portuguese General Data Protection Regulation, but are available from the corresponding author on reasonable request.

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
