# Peer review of "Clostridioides difficile Infection in Hospitalized Patients—A Retrospective Epidemiological Study"

_healthcare, 2023, doi:10.3390/healthcare12010076_

Round 1
Reviewer 1 Report
Comments and Suggestions for Authors
The article is interesting and well-written. Methods are correctly described, and the sample is big enough.
The only thing I suggest is to enrich the introduction and the conclusions talking about the effects and the costs of healthcare-related forms of CDI.
To Enrich the paper i could suggest the following citation:
Medico-Legal Aspects of Hospital-Acquired Infections: 5-Years of Judgements of the Civil Court of Rome
Healthcare
Treglia M et al.
Author Response
Dear Reviewer,
We analized all your comments as per the following:
Response to Reviewer 1 Comments
Point-by-point response to Comments and Suggestions for Authors
Comments 1: The article is interesting and well-written. Methods are correctly described, and the sample is big enough.
Response 1: Your kind words about the article being interesting and well-written, along with acknowledging the accurate description of our methods and the adequacy of our sample size, are deeply appreciated. Your positive assessment serves as a testament to our team's dedication and effort in conducting this research. We sincerely appreciate your efforts and look forward to any further insights you may have.
Comment 2: The only thing I suggest is to enrich the introduction and the conclusions talking about the effects and the costs of healthcare-related forms of CDI.
Response 2: Thank you for bringing this to our attention. We enhanced our introduction to ensure it contains ample background information and incorporates all pertinent references based on your input. Consequently, we will include this suggestion in our study's introduction section to strengthen the robustness of our conclusions.
Comments 3: To Enrich the paper i could suggest the following citation: Medico-Legal Aspects of Hospital-Acquired Infections: 5-Years of Judgements of the Civil Court of Rome.
Response 3: The authors have unanimously agreed with the suggestion. Subsequently, we have revised the content in line with your suggestion to emphasize this point. All authors concur that this inclusion is a positive aspect for our paper. Therefore, we will integrate your suggestion into the discussion section of our article.
Please see the attachment file.

Reviewer 2 Report
Comments and Suggestions for Authors
In the present manuscript, the authors carried out a single-center, observational and retrospective study in which they analyzed the incidence of Clostridioides difficile infection and described the severity and outcomes of this event in patients hospitalized at the Espírito Santo Hospital in Évora, Portugal, during 2018. Despite the limitations of the study detailed in the manuscript, the article makes a certain contribution to the existing literature, highlighting the exponential increase in the incidence of infections in the aforementioned hospital, particularly episodes acquired in the community.
Some comments:
Acronyms/abbreviations/initialisms should be defined the first time they appear in each of three sections: the abstract; the main text; the first figure or table, and then abbreviated on all other occasions.
Keywords: it is advisable to use terms different from those provided in the title. Please change “Clostridioides difficile infection (CDI)”.
Line 34: “It is responsible for Clostridioides difficile infection…”. Please rewrite this. Also, once the species is named, use the abbreviated genus in subsequent mentions (see also line 605).
Line 80: please delete one “22”.
Lines 102-103: “other strategies…effective drugs”. References are needed.
Line 117: “toxin” change to “toxins”.
Line 154: the authors must describe the steps taken to carry out their systematic literature review (databases used, inclusion and exclusion criteria…).
Please check the consistency of the data presented in Table 2 and those described in lines 259-271.
Lines 276-277: “most patients presented…serum creatinine levels and above-normal serum lactate level”. However, 50.0 % (high creatinine) and 46.7% (hyperlactatemia) are not the majority.
Lines 302-303: “There were statistically significant correlations between serum lactate levels and CCI p=0,023”. What implications could this result have?
Please check the consistency of the data presented in Table 3 and those described in lines 306-308: “All sCDI and rCDI episodes had previous antimicrobial consumption. Moreover, only one patient with a non-severe episode and a CA-CDI had no record of previous antibiotic consumption”.
Please check the data in Table 3, specially those corresponding to “Group age of participant” and “Antimicrobial consumption”.
Please check the consistency of the data presented in Table 4 and those described in line 351: “Furthermore, 88% of all sCDI cases had cephalosporin consumption”.
Lines 402-403: “Type of Initial Therapy vs Sex (p = 0.047)”. Results and discussion on this are lacking.
In this reviewer’s opinion, Table 7 and Figure 1 should be included in the results section.
What reasons could explain the results obtained by Barbosa-Martins et al. and Teixeira et al.?
Lines 555-556: “until 2019 followed by a plateau”. Please rewrite this as shown in Figure 1.
In this reviewer’s opinion, section 4.6 should be merged with 4.5.
References should be formatted according to the instruction for authors.
Author Response
Dear Reviewer,
We analized all your comments as per the following:
Point-by-point response to Comments and Suggestions for Authors
Comments 1: Acronyms/abbreviations/initialisms should be defined the first time they appear in each of three sections: the abstract; the main text; the first figure or table, and then abbreviated on all other occasions.
Response 1: Thank you for bringing this to our attention. The authors concur with this observation. As a result, we have rectified all the acronyms/abbreviations/initialisms present in the main text.
Comments 2: Keywords: it is advisable to use terms different from those provided in the title. Please change “Clostridioides difficile infection (CDI)”.
Response 2: The authors concur. We have consequently taken steps to modify and emphasize this aspect. However, it's important to note that the journal's regulations do not explicitly forbid this specific keyword definition. As a result, we will rephrase it using solely "Clostridioides difficile infection."
As described on MDPI Instructions for Authors: “Keywords: Three to ten pertinent keywords need to be added after the abstract. We recommend that the keywords are specific to the article, yet reasonably common within the subject discipline”.
Comments 3: Line 34: “It is responsible for Clostridioides difficile infection…”. Please rewrite this. Also, once the species is named, use the abbreviated genus in subsequent mentions (see also line 605).
Response 3: Thank you for bringing this to our attention. The authors are grateful for your comment and review, and concur with the suggestion. As a result, we have updated all instances where the species is named by using the abbreviated genus.
Comments 4: Line 80: please delete one “22”.
Response 4: Thank you for bringing this to our attention. The author overlooked this gap, which has now been addressed and updated in the text of the reviewed manuscript.
Comments 5: Lines 102-103: “other strategies…effective drugs”. References are needed.
Response 5: We appreciate you highlighting this matter to us. The authors have already updated the bibliographic references in the reviewed manuscript.
Comments 6: Line 117: “toxin” change to “toxins”.
Response 6: The authors concur with the comment. The term “toxins” have been updated accordingly.
Comments 7: Line 154: the authors must describe the steps taken to carry out their systematic literature review (databases used, inclusion and exclusion criteria…).
Response 7: Thank you for highlighting this. The authors inadvertently made an error. Instead of conducting a systematic review of the literature, our approach involved a descriptive review. This was carried out specifically to perform a comparative analysis between the epidemiological data obtained in our research and data from other cohorts and investigations from Portugal and other countries. These studies met similar eligibility criteria and were utilized for evaluating CDI epidemiological overall evolution and the consistency with our results. This epidemiological descriptive review also included a comparison between cases of community-acquired (CA-CDI) and hospital-acquired (HA-CDI) infections at Hospital do Espírito Santo in Évora, during 2000-2008 by Silva et al. 2012, over a period of 22 years.
Comments 8: Please check the consistency of the data presented in Table 2 and those described in lines 259-271.
Response 8: The data presented in Table 2 has been rectified, and the number of Healthcare and Community-CDI cases are now represented in 'n' format in the text session. This adjustment aims to enhance the visual comprehension of healthcare and community-acquired cases for better perceptibility.
Comments 9: Lines 276-277: “most patients presented…serum creatinine levels and above-normal serum lactate level”. However, 50.0 % (high creatinine) and 46.7% (hyperlactatemia) are not the majority.
Response 9: Thank you for bringing this to our attention. It was erroneously mentioned that most patients presented severe analytical complications, which was not the intended statement. The current manuscript has been revised and updated to accurately reflect this information. We adjusted for the fact that most cases of sCDI showed elevated levels of serum creatinine and above-normal serum lactate levels.
Comments 10: Lines 302-303: “There were statistically significant correlations between serum lactate levels and CCI p=0,023”. What implications could this result have?
Response 10: The implications have been included in the new manuscript's discussion section. These statistically implications are also supported by referenced literature for further substantiation, namely:
[53] Kim, S.G., Lee, J., Yun, D., Kang, M.W., Kim, Y.C., Kim, D.K., ... Han, S.S. (2023). Hyperlactatemia is a predictor of mortality in patients undergoing continuous renal replacement therapy for acute kidney injury. BMC Nephrology, 24(1), 1–8. https://doi.org/10.1186/s12882-023-03063-y
[54] Silva, C.M., Baptista, J.P., Mergulhão, P., Froes, F., Gonçalves-Pereira, J., Pereira, J.M., ... Paiva, J.A. (2022). Prognostic value of hyperlactatemia in infected patients admitted to intensive care units: a multicenter study. Revista Brasileira de Terapia Intensiva, 34(1), 154–162. https://doi.org/10.5935/0103-507x.20220010-en
Comments 11: Please check the consistency of the data presented in Table 3 and those described in lines 306-308: “All sCDI and rCDI episodes had previous antimicrobial consumption. Moreover, only one patient with a non-severe episode and a CA-CDI had no record of previous antibiotic consumption”.
Response 11: Thank you for addressing this matter. An error occurred during the data transcription while formatting our research to comply with the MDPI format, leading to inaccuracies in the table presentation. To resolve this issue, the data has been carefully reviewed and corrected based on our study results. The revised data now aligns with the information presented in the text section, rectifying the inaccuracies.
Comments 12: Please check the data in Table 3, specially those corresponding to “Group age of participant” and “Antimicrobial consumption”.
Response 12: As in Comment 11, there was a mistake in data transcription when adapting our research to adhere to the MDPI format, resulting in inaccuracies in the table presentation. To rectify this, we meticulously reviewed and corrected the data using our study results. The updated data now corresponds accurately with the information provided in the text, resolving the inaccuracies.
Comments 13: Please check the consistency of the data presented in Table 4 and those described in line 351: “Furthermore, 88% of all sCDI cases had cephalosporin consumption”.
Response 13: Thank you for highlighting this. The alignment between the data presented in Table 4 and the information described in line 351 as "Furthermore, 88% of all sCDI cases had cephalosporin consumption" is inaccurate. Therefore, we will revise this text section accordingly to ensure accuracy and consistency.
Comments 14: Lines 402-403: “Type of Initial Therapy vs Sex (p = 0.047)”. Results and discussion on this are lacking.
Response 14: The authors have decided to remove this section; however, some bibliographic references have been identified that could explain this matter. Most clinical guidelines recommended in 2018 Metronidazole and/or Vancomycin, as first-line treatments for CDI, irrespective of any history of liver failure or gender. This guideline-based approach likely contributes to the observed strong correlations between the choice of initial therapy and gender, within our study sample, as described by (Dotson et al., 2018) (Trifan et al., 2015).
Comments 15: In this reviewer’s opinion, Table 7 and Figure 1 should be included in the results section.
Response 15: The authors appreciate the suggestion. We have reached a consensus and opted to convert and incorporate Table 7 and Figure 1 into the results section as it aligns better with the context and flow of the manuscript. It is now updated.
Comments 16: What reasons could explain the results obtained by Barbosa-Martins et al. and Teixeira et al.?
Response 16: We cannot extrapolate or make definitive conclusions based on other studies, but the hospitals where these studies were conducted might have implemented superior hygiene measures and antimicrobial prescription programs, which were not explored in the studies by Barbosa-Martins et al., 2018 and Teixeira et al., 2017. Additionally, the inclusion criteria might differ, potentially being more stringent. These hospitals could be smaller and categorized according to DL ordinance 82/2014, which classifies services and establishments within the National Health Service. The total number of inpatients admitted during the study period, and subsequently included in the study, was larger, involving multicenter retrospective study techniques.
Comments 17: Lines 555-556: “until 2019 followed by a plateau”. Please rewrite this as shown in Figure 1.
Response 17: Lines 555-556: "until 2019, which was subsequently revised from 'plateau' to 'decrease' for accuracy. The authors appreciate this correction, and the updated manuscript reflects this amendment consistently, as the following: “Our documented data suggests a significant increase in overall CDI incidence until 2019 followed by a considerable decrease thereafter.”
Comments 18: In this reviewer’s opinion, section 4.6 should be merged with 4.5.
Response 18: We are immensely grateful for the reviewer's opinion. However, in this specific context, the authors hold the viewpoint that these are two distinct topics that complement each other effectively. While we acknowledge the suggestion, we believe that the pandemic period should be treated separately from the epidemiology of CDI in HESE (Hospital Espírito Santo de Évora).
Comments 19: References should be formatted according to the instruction for authors.
Response 19: The bibliographic references have been revised and formatted according to the specific guidelines and formatting style required for MDPI articles within the updated manuscript.
Please see the attachment file.

Reviewer 3 Report
Comments and Suggestions for Authors
The manuscript titled "Clostridioides difficile Infection in Hospitalized Patients: A Retrospective Epidemiological Study" provides valuable insights into the epidemiology and management of CDI in a hospital setting. The primary objective was to examine the epidemiological characteristics, including incidence, severity, and outcomes, of CDI in a hospital context, highlighting the changing trends in the global epidemiology of this infection. This research provides vital insights into the epidemiology and management of CDI, emphasizing the evolving nature of its incidence and severity, the crucial role of antibiotic stewardship, and the potential of probiotics in enhancing treatment outcomes. Overall, this manuscript is well-structured and provides valuable information on the epidemiological characteristics, treatment outcomes, and associated factors of CDI while revealing the trends in CDI incidence. However, there are a few areas that require attention and improvement before publication.
Major issues
1. The sample size is too small. The study included only 30 patients out of 395 for further analysis. The small sample size may limit the generalizability of the findings to a wider population. And this study was conducted in a specific hospital located in Portugal, the finding may not be representative. The authors should include more hospitals and thus more samples for analysis.
2. The study is mainly based on the data from year 2018. The authors need to clarify why year 2018 was selected rather than utilizing more recent years as the authors have included the data from year 2019-2022.
3. Section 3.2: Please summarize the data into a tabular format for clarity, like the number of patients in HA-CDI/CA-CDI, age group, gender, and detection methods.
4. Table 2: How the percentages were calculated in table 2? It seems that the numbers are not consistent with the interpretation in line 259-267.
Minor issues
1. Line 82: abbreviation “SoC” should be clarified in its first appearance.
2. Line 581-582: Please indicate the reference data. And why September but not December?
3. Line 589-591: Please indicate the reference data.
Author Response
Point-by-point response to Comments and Suggestions for Authors
Comments 1: The sample size is too small. The study included only 30 patients out of 395 for further analysis. The small sample size may limit the generalizability of the findings to a wider population. And this study was conducted in a specific hospital located in Portugal, the finding may not be representative. The authors should include more hospitals and thus more samples for analysis.
Response 1: Thank you for bringing this to our attention, we appreciate your constructive criticism and assure you that we have carefully considered your suggestions in revising the manuscript. We agree with this comment. The inclusion of 30 patients in the 2018 study was intentional, aiming for a more comprehensive analysis, as it was a retrospective study conducted within a single center. Collecting data from other hospitals posed greater challenges, requiring strict adherence to similar patterns and eligibility criteria. This suggestion of yours aligns perfectly with our intentions for future research directions, and we are committed to pursuing this avenue to enrich the depth and breadth of our study.
Comments 2: The study is mainly based on the data from year 2018. The authors need to clarify why year 2018 was selected rather than utilizing more recent years as the authors have included the data from year 2019-2022.
Response 2: The authors have duly acknowledged and implemented revisions to underscore this point. The actual sample consists of 1450 suspected CD infections spanning from 2018 to 2022, of which 134 met the eligibility criteria and were selected for the epidemiological investigation. Limitations within the study address this issue. Moreover, we aim to conduct a more recent comprehensive data analysis covering a five-year period, particularly focusing on healthcare and community-acquired cases. This analysis will compare data from the same hospital institution and other cohorts in Portugal and elsewhere, aligning with similar eligibility criteria to ensure a robust analysis of the existing literature.
Comments 3: Section 3.2: Please summarize the data into a tabular format for clarity, like the number of patients in HA-CDI/CA-CDI, age group, gender, and detection methods.
Response 3: The authors concur with the comment. We opted not to include the results in a table to prevent adding another table. Hence, we chose to integrate summarized information into the text within the Results section. This approach was chosen since the details regarding Healthcare and Community-Acquired CDI cases were elaborated in table 2, and the age group information is already presented in table 3. If needed, we will update the manuscript by providing this information in a table format. The reviewer recommendation is to include these results in a table, here is a draft of the proposed table for your consideration.
Table 1. Patient’s sociodemographic characteristics, infection acquisition site and detection methods.
|
Total Sample Size n=30 |
No. (%) |
|
|
Sex of participants |
Female |
16 (53.3) |
|
Male |
14 (46.7) |
|
|
Group age of participants |
< 65 years old |
5 (16.7) |
|
65-80 years old |
11 (36.7) |
|
|
≥ 80 years old |
14 (46.7) |
|
|
Acquisition Site of CD |
Healthcare-acquired CDI |
19 (63.3) |
|
Community-acquired CDI |
11 (36.7) |
|
|
Detection Methods |
Detection of CD Toxins A and B |
28 (93.3) |
|
Histological examination |
2 (6.7) |
|
Comments 4: Table 2: How the percentages were calculated in table 2? It seems that the numbers are not consistent with the interpretation in line 259-267.
Response 4: The data presented in Table 2 has been rectified, and the number of cases is now represented in 'n' format. This adjustment aims to enhance the visual perception of healthcare and community-acquired cases for better understanding.
Comments 5: Line 82: abbreviation “SoC” should be clarified in its first appearance.
Response 5: Thank you for acknowledging and suggesting improvements. The abbreviations have been clarified prior to their appearance in the manuscript.
Comments 6: Line 581-582: Please indicate the reference data. And why September but not December?
Response 6: Thank you for your information. The data presented in this study reflects information available up until September, as sourced from reference [63]. This cut-off date predates December, as indicated in the subsequent reference provided by Vendrik et al. (2022). Their study titled "Comparison of trends in Clostridioides difficile infections in hospitalised patients during the first and second waves of the COVID-19 pandemic: A retrospective sentinel surveillance study" offers more recent insights.
Comments 7: Line 589-591: Please indicate the reference data.
Response 7: Thank you for bringing this to attention: The reference data has already been included in the updated manuscript.
Please see the attachment document.

Round 2
Reviewer 2 Report
Comments and Suggestions for Authors
The authors have addressed all my concerns.
Reviewer 3 Report
Comments and Suggestions for Authors
The authors have issued my concerns.